# Contrastive Learning for Image Captioning

**Bo Dai**          **Dahua Lin**
Department of Information Engineering, The Chinese University of Hong Kong
db014@ie.cuhk.edu.hk    dhlin@ie.cuhk.edu.hk

## Abstract

Image captioning, a popular topic in computer vision, has achieved substantial progress in recent years. However, the *distinctiveness* of natural descriptions is often overlooked in previous work. It is closely related to the quality of captions, as distinctive captions are more likely to describe images with their unique aspects. In this work, we propose a new learning method, Contrastive Learning (CL), for image captioning. Specifically, via two constraints formulated on top of a reference model, the proposed method can encourage distinctiveness, while maintaining the overall quality of the generated captions. We tested our method on two challenging datasets, where it improves the baseline model by significant margins. We also showed in our studies that the proposed method is generic and can be used for models with various structures.

## 1   Introduction

Image captioning, a task to generate natural descriptions of images, has been an active research topic in computer vision and machine learning. Thanks to the advances in deep neural networks, especially the wide adoption of RNN and LSTM, there has been substantial progress on this topic in recent years [23, 24, 15, 19]. However, studies [1, 3, 2, 10] have shown that even the captions generated by state-of-the-art models still leave a lot to be desired. Compared to human descriptions, machine-generated captions are often quite rigid and tend to favor a *"safe"* (*i.e.* matching parts of the training captions in a word-by-word manner) but *restrictive* way. As a consequence, captions generated for different images, especially those that contain objects of the same categories, are sometimes very similar [1], despite their differences in other aspects.

We argue that **distinctiveness**, a property often overlooked in previous work, is significant in natural language descriptions. To be more specific, when people describe an image, they often mention or even emphasize the *distinctive* aspects of an image that distinguish it from others. With a distinctive description, someone can easily identify the image it is referring to, among a number of similar images. In this work, we performed a *self-retrieval* study (see Section 4.1), which reveals the lack of distinctiveness affects the quality of descriptions.

From a technical standpoint, the lack of *distinctiveness* is partly related to the way that the captioning model was learned. A majority of image captioning models are learned by Maximum Likelihood Estimation (MLE), where the probabilities of training captions conditioned on corresponding images are maximized. While well grounded in statistics, this approach does not explicitly promote distinctiveness. Specifically, the differences among the captions of different images are not explicitly taken into account. We found empirically that the resultant captions highly resemble the training set in a word-by-word manner, but are not *distinctive*.

In this paper, we propose **Contrastive Learning (CL)**, a new learning method for image captioning, which explicitly encourages *distinctiveness*, while maintaining the overall quality of the generated captions. Specifically, it employs a baseline, *e.g.* a state-of-the-art model, as a *reference*. During learning, in addition to true image-caption pairs, denoted as $(I, c)$, this method also takes as input

*mismatched pairs*, denoted as $(I, c_/)$, where $c_/$ is a caption describing another image. Then, the target model is learned to meet two goals, namely (1) giving higher probabilities $p(c|I)$ to positive pairs, and (2) lower probabilities $p(c_/|I)$ to negative pairs, compared to the reference model. The former ensures that the overall performance of the target model is not inferior to the reference; while the latter encourages distinctiveness.

It is noteworthy that the proposed learning method (CL) is generic. While in this paper, we focused on models based on recurrent neural networks [23, 15], the proposed method can also generalize well to models based on other formulations, *e.g.* probabilistic graphical models [4, 9]. Also, by choosing the state-of-the-art model as the reference model in CL, one can build on top of the latest advancement in image captioning to obtain improved performances.

## 2   Related Work

**Models for Image Captioning**   The history of image captioning can date back to decades ago. Early attempts are mostly based on detections, which first detect visual concepts (*e.g.* objects and their attributes) [9, 4] followed by template filling [9] or nearest neighbor retrieving for caption generation [2, 4]. With the development of neural networks, a more powerful paradigm, *encoder-and-decoder*, was proposed by [23], which then becomes the core of most state-of-the-art image captioning models. It uses a CNN [20] to represent the input image with a feature vector, and applies a LSTM net [6] upon the feature to generate words one by one.

Based on the encoder-and-decoder, many variants are proposed, where *attention mechanism* [24] appears to be the most effective add-on. Specifically, attention mechanism replaces the feature vector with a set of feature vectors, such as the features from different regions [24] , and those under different conditions [27]. It also uses the LSTM net to generate words one by one, where the difference is that at each step, a mixed guiding feature over the whole feature set, will be *dynamically* computed. In recent years, there are also approaches combining attention mechanism and detection. Instead of doing attention on features, they consider the attention on a set of detected visual concepts, such as attributes [25] and objects [26].

Despite of the specific structure of any image captioning model, it is able to give $p(c|I)$, the probability of a caption conditioned on an image. Therefore, all image captioning models can be used as the target or the reference in CL method.

**Learning Methods for Image Captioning**   Many state-of-the-art image captioning models adopt *Maximum Likelihood Estimation (MLE)* as their learning method, which maximizes the conditional log-likelihood of the training samples, as:

$$\sum_{(c_i, I_i) \in \mathcal{D}} \sum_{t=1}^{T_i} \ln p(w_i^{(t)} | I_i, w_i^{(t-1)}, ..., w_i^{(1)}, \boldsymbol{\theta}), \tag{1}$$

where $\boldsymbol{\theta}$ is the parameter vector, $I_i$ and $c_i = (w_i^{(1)}, w_i^{(2)}, ..., w_i^{(T_i)})$ are a training image and its caption. Although effective, some issues, including high resemblance in model-gerenated captions, are observed [1] on models learned by MLE.

Facing these issues, alternative learning methods are proposed in recent years. Techniques of reinforcement learning (RL) have been introduced in image captioning by [19] and [14]. RL sees the procedure of caption generation as a procedure of sequentially sampling actions (words) in a policy space (vocabulary). The rewards in RL are defined to be evaluation scores of sampled captions. Note that distinctiveness has not been considered in both approaches, RL and MLE.

Prior to this work, some relevant ideas have been explored [21, 16, 1]. Specifically, [21, 16] proposed an introspective learning (IL) approach that learns the target model by comparing its outputs on $(I, c)$ and $(I_/, c)$. Note that IL uses the target model itself as a reference. On the contrary, the reference model in CL provides more *independent* and *stable* indications about distinctiveness. In addition, $(I_/, c)$ in IL is pre-defined and fixed across the learning procedure, while the negative sample in CL, *i.e.* $(I, c_/)$, is *dynamically* sampled, making it more diverse and random. Recently, Generative Adversarial Networks (GAN) was also adopted for image captioning [1], which involves an evaluator that may help promote the distinctiveness. However, this evaluator is *learned* to *directly* measure the

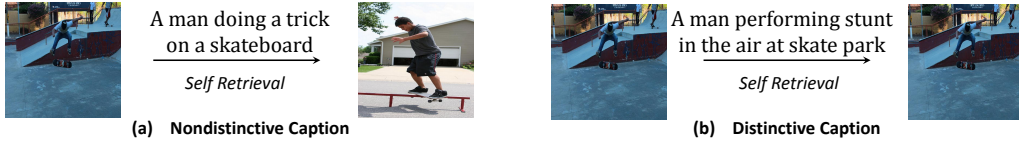

| | | A man doing a trick on a skateboard | | | A man performing stunt in the air at skate park | |
|---|---|---|---|---|---|---|

(a)  **Nondistinctive Caption**                              (b)  **Distinctive Caption**

Figure 1: This figure illustrates respectively a nondistinctive and distinctive captions of an image, where the nondistinctive one fails to retrieve back the original image in *self retrieval* task.

|  | Self Retrieval Top-K Recall | | | | Captioning | |
|---|---|---|---|---|---|---|
| Method | 1 | 5 | 50 | 500 | ROUGE_L | CIDEr |
| Neuraltalk2 [8] | 0.02 | 0.32 | 3.02 | 27.50 | 0.652 | 0.827 |
| AdaptiveAttention [15] | 0.10 | 0.96 | 11.76 | 78.46 | 0.689 | 1.004 |
| AdaptiveAttention + CL | 0.32 | 1.18 | 11.84 | 80.96 | 0.695 | 1.029 |

Table 1: This table lists results of self retrieval and captioning of different models. The results are reported on standard MSCOCO test set. See sec 4.1 for more details.

distinctiveness as a parameterized approximation, and the approximation accuracy is not ensured in GAN. In CL, the *fixed* reference provides stable *bounds* about the distinctiveness, and the bounds are supported by the model's performance on image captioning. Besides that, [1] is specifically designed for models that generate captions word-by-word, while CL is more generic.

## 3  Background

Our formulation is partly inspired by *Noise Contrastive Estimation (NCE)* [5]. NCE is originally introduced for estimating probability distributions, where the partition functions can be difficult or even infeasible to compute. To estimate a parametric distribution $p_m(.;\boldsymbol{\theta})$, which we refer to as the *target* distribution, NCE employs not only the observed samples $X = (\mathbf{x}_1, \mathbf{x}_2, ..., \mathbf{x}_{T_m})$, but also the samples drawn from a *reference* distribution $p_n$, denoted as $Y = (\mathbf{y}_1, \mathbf{y}_2, ..., \mathbf{y}_{T_n})$. Instead of estimating $p_m(.;\boldsymbol{\theta})$ directly, NCE estimates the density ratio $p_m/p_n$ by training a classifier based on logistic regression.

Specifically, let $U = (\mathbf{u}_1, ..., \mathbf{u}_{T_m+T_n})$ be the union of $X$ and $Y$. A binary class label $C_t$ is assigned to each $u_t$, where $C_t = 1$ if $u_t \in X$ and $C_t = 0$ if $u_t \in Y$. The posterior probabilities for the class labels are therefore

$$P(C = 1|\mathbf{u}, \boldsymbol{\theta}) = \frac{p_m(\mathbf{u};\boldsymbol{\theta})}{p_m(\mathbf{u};\boldsymbol{\theta}) + \nu p_n(\mathbf{u})}, \qquad P(C = 0|\mathbf{u}, \boldsymbol{\theta}) = \frac{\nu p_n(\mathbf{u})}{p_m(\mathbf{u};\boldsymbol{\theta}) + \nu p_n(\mathbf{u})}, \qquad (2)$$

where $\nu = T_n/T_m$. Let $G(\mathbf{u};\boldsymbol{\theta}) = \ln p_m(\mathbf{u};\boldsymbol{\theta}) - \ln p_n(\mathbf{u})$ and $h(\mathbf{u}, \boldsymbol{\theta}) = P(C = 1|\mathbf{u}, \boldsymbol{\theta})$, then we can write

$$h(\mathbf{u};\boldsymbol{\theta}) = r_\nu(G(\mathbf{u};\boldsymbol{\theta})), \quad \text{with} \quad r_\nu(z) = \frac{1}{1 + \nu \exp(-z)}. \qquad (3)$$

The objective function of NCE is the joint conditional log-probabilities of $C_t$ given the samples $U$, which can be written as

$$\mathcal{L}(\boldsymbol{\theta}; X, Y) = \sum_{t=1}^{T_m} \ln[h(\mathbf{x}_t;\boldsymbol{\theta})] + \sum_{t=1}^{T_n} \ln[1 - h(\mathbf{y}_t;\boldsymbol{\theta})]. \qquad (4)$$

Maximizing this objective with respect to $\boldsymbol{\theta}$ leads to an estimation of $G(\cdot;\boldsymbol{\theta})$, the logarithm of the density ratio $p_m/p_n$. As $p_n$ is a known distribution, $p_m(:|\boldsymbol{\theta})$ can be readily derived.

## 4  Contrastive Learning for Image Captioning

Learning a model by characterizing desired properties relative to a strong baseline is a convenient and often quite effective way in situations where it is hard to describe these properties directly. Specifically, in image captioning, it is difficult to characterize the distinctiveness of natural image descriptions via a set of rules, without running into the risk that some subtle but significant points are

missed. Our idea in this work is to introduce a baseline model as a reference, and try to enhance the distinctiveness on top, while maintaining the overall quality of the generated captions.

In the following we will first present an empirical study on the correlation between *distinctiveness* of its generated captions and the *overall performance* of a captioning model. Subsequently, we introduce the main framework of *Contrastive Learning* in detail.

## 4.1 Empirical Study: Self Retrieval

In most of the existing learning methods of image captioning, models are asked to generate a caption that best describes the semantics of a given image. In the meantime, **distinctiveness** of the caption, which, on the other hand, requires the image to be the best matching *among all images* for the caption, has not been explored. However, distinctiveness is crucial for high-quality captions. A study by Jas [7] showed that *specificity* is common in human descriptions, which implies that image descriptions often involve distinctive aspects. Intuitively, a caption satisfying this property is very likely to contain key and unique content of the image, so that the original image could easily be retrieved when the caption is presented.

To verify this intuition, we conducted an empirical study which we refer to as *self retrieval*. In this experiment, we try to retrieve the original image given its model-generated caption and investigate top-$k$ recalls, as illustrated in Figure 1. Specifically, we randomly sampled $5,000$ images $(I_1, I_2, ..., I_{5000})$ from standard MSCOCO [13] test set as the experiment benchmark. For an image captioning model $p_m(:, \boldsymbol{\theta})$, we first ran it on the benchmark to get corresponding captions $(c_1, c_2, ..., c_{5000})$ for the images. After that, using each caption $c_t$ as a query, we computed the conditional probabilities $(p_m(c_t|I_1), p_m(c_t|I_2), ..., p_m(c_t|I_{5000}))$, which were used to get a ranked list of images, denoted by $\mathbf{r}_t$. Based on all ranked lists, we can compute top-$k$ recalls, which is the fraction of images within top-$k$ positions of their corresponding ranked lists. The top-$k$ recalls are good indicators of how well a model captures the distinctiveness of descriptions.

In this experiment, we compared three different models, including *Neuraltalk2* [8] and *AdaptiveAttention* [15] that are learned by MLE, as well as *AdaptiveAttention* learned by our method. The top-$k$ recalls are listed in Table 1, along with overall performances of these models in terms of *Rouge* [12] and *Cider* [22]. These results clearly show that the recalls of self retrieval are positively correlated to the performances of image captioning models in classical captioning metrics. Although most of the models are not explicitly learned to promote distinctiveness, the one with better recalls of self retrieval, which means the generated-captions are more distinctive, performs better in the image captioning evaluation. Such positive correlation clearly demonstrates the significance of *distinctiveness* to captioning performance.

## 4.2 Contrastive Learning

In Contrastive Learning (CL), we learn a target image captioning model $p_m(::; \boldsymbol{\theta})$ with parameter $\boldsymbol{\theta}$ by constraining its behaviors relative to a reference model $p_n(::; \boldsymbol{\phi})$ with parameter $\boldsymbol{\phi}$. The learning procedure requires two sets of data: (1) the observed data $X$, which is a set of ground-truth image-caption pairs $((c_1, I_1), (c_2, I_2), ..., (c_{T_m}, I_{T_m}))$, and is readily available in any image captioning dataset, (2) the noise set $Y$, which contains mismatched pairs $((c_{/1}, I_1), (c_{/2}, I_2), ..., (c_{/T_n}, I_{T_n}))$, and can be generated by randomly sampling $c_{/t} \in \mathcal{C}_{/I_t}$ for each image $I_t$, where $\mathcal{C}_{/I_t}$ is the set of all ground-truth captions except captions of image $I_t$. We refer to $X$ as *positive pairs* while $Y$ as *negative pairs*.

For any pair $(c, I)$, the target model and the reference model will respectively give their estimated conditional probabilities $p_m(c|I, \boldsymbol{\theta})$ and $p_n(c|I, \boldsymbol{\phi})$. We wish that $p_m(c_t|I_t, \boldsymbol{\theta})$ is greater than $p_n(c_t|I_t, \boldsymbol{\phi})$ for any positive pair $(c_t, I_t)$, and vice versa for any negative pair $(c_{/t}, I_t)$. Following this intuition, our initial attempt was to define $D((c, I); \boldsymbol{\theta}, \boldsymbol{\phi})$, the difference between $p_m(c|I, \boldsymbol{\theta})$ and $p_n(c|I, \boldsymbol{\phi})$, as

$$D((c, I); \boldsymbol{\theta}, \boldsymbol{\phi}) = p_m(c|I, \boldsymbol{\theta}) - p_n(c|I, \boldsymbol{\phi}), \tag{5}$$

and set the loss function to be:

$$\mathcal{L}'(\boldsymbol{\theta}; X, Y, \boldsymbol{\phi}) = \sum_{t=1}^{T_m} D((c_t, I_t); \boldsymbol{\theta}, \boldsymbol{\phi}) - \sum_{t=1}^{T_n} D((c_{/t}, I_t); \boldsymbol{\theta}, \boldsymbol{\phi}). \tag{6}$$

In practice, this formulation would meet with several difficulties. First, $p_m(c|I, \boldsymbol{\theta})$ and $p_n(c|I, \boldsymbol{\phi})$ are very small ($\sim 1e\text{-}8$), which may result in numerical problems. Second, Eq (6) treats easy samples, hard samples, and mistaken samples equally. This, however, is not the most effective way. For example, when $D((c_t, I_t); \boldsymbol{\theta}, \boldsymbol{\phi}) \gg 0$ for some positive pair, further increasing $D((c_t, I_t); \boldsymbol{\theta}, \boldsymbol{\phi})$ is probably not as effective as updating $D((c_{t'}, I_{t'}); \boldsymbol{\theta}, \boldsymbol{\phi})$ for another positive pair, for which $D((c_{t'}, I_{t'}); \boldsymbol{\theta}, \boldsymbol{\phi})$ is much smaller.

To resolve these issues, we adopted an alternative formulation inspired by NCE (sec 3), where we replace the difference function $D((c, I); \boldsymbol{\theta}, \boldsymbol{\phi})$ with a log-ratio function $G((c, I); \boldsymbol{\theta}, \boldsymbol{\phi})$:

$$G((c, I); \boldsymbol{\theta}, \boldsymbol{\phi}) = \ln p_m(c|I, \boldsymbol{\theta}) - \ln p_n(c|I, \boldsymbol{\phi}), \tag{7}$$

and further use a logistic function $r_\nu$ (Eq(3)) after $G((c, I); \boldsymbol{\theta}, \boldsymbol{\phi})$ to saturate the influence of easy samples. Following the notations in NCE, we let $\nu = T_n/T_m$, and turn $D((c, I); \boldsymbol{\theta}, \boldsymbol{\phi})$ into:

$$h((c, I); \boldsymbol{\theta}, \boldsymbol{\phi}) = r_\nu(G((c, I); \boldsymbol{\theta}, \boldsymbol{\phi}))). \tag{8}$$

Note that $h((c, I); \boldsymbol{\theta}, \boldsymbol{\phi}) \in (0, 1)$. Then, we define our updated loss function as:

$$\mathcal{L}(\boldsymbol{\theta}; X, Y, \boldsymbol{\phi}) = \sum_{t=1}^{T_m} \ln[h((c_t, I_t); \boldsymbol{\theta}, \boldsymbol{\phi})] + \sum_{t=1}^{T_n} \ln[1 - h((c_{/t}, I_t); \boldsymbol{\theta}, \boldsymbol{\phi})]. \tag{9}$$

For the setting of $\nu = T_n/T_m$, we choose $\nu = 1$, *i.e.* $T_n = T_m$, to ensure balanced influences from both positive and negative pairs. This setting consistently yields good performance in our experiments. Furthermore, we copy $X$ for $K$ times and sample $K$ different $Y$s, in order to involve more diverse negative pairs without overfitted to them. In practice we found $K = 5$ is sufficient to make the learning stable. Finally, our objective function is defined to be

$$J(\boldsymbol{\theta}) = \frac{1}{K} \frac{1}{T_m} \sum_{k=1}^{K} \mathcal{L}(\boldsymbol{\theta}; X, Y_k, \boldsymbol{\phi}). \tag{10}$$

Note that $J(\boldsymbol{\theta})$ attains its upper bound $0$ if positive and negative pairs can be perfectly distinguished, namely, for all $t$, $h((c_t, I_t); \boldsymbol{\theta}, \boldsymbol{\phi}) = 1$ and $h((c_{/t}, I_t); \boldsymbol{\theta}, \boldsymbol{\phi}) = 0$. In this case, $G((c_t, I_t); \boldsymbol{\theta}, \boldsymbol{\phi}) \to \infty$ and $G((c_{/t}, I_t); \boldsymbol{\theta}, \boldsymbol{\phi}) \to -\infty$, which indicates the target model will give higher probability $p(c_t|I_t)$ and lower probability $p(c_{/t}|I_t)$, compared to the reference model. Towards this goal, the learning process would encourage *distinctiveness* by suppressing negative pairs, while maintaining the overall performance by maximizing the probability values on positive pairs.

## 4.3 Discussion

Maximum Likelihood Estimation (MLE) is a popular learning method in the area of image captioning [23, 24, 15]. The objective of MLE is to maximize *only* the probabilities of ground-truth image-caption pairs, which may lead to some issues [1], including high resemblance in generated captions. While in CL, the probabilities of ground-truth pairs are *indirectly* ensured by the positive constraint (the first term in Eq(9)), and the negative constraint (the second term in Eq(9)) suppresses the probabilities of mismatched pairs, forcing the target model to also learn from distinctiveness.

Generative Adversarial Network (GAN) [1] is a similar learning method that involves an auxiliary model. However, in GAN the auxiliary model and the target model follow two *opposite* goals, while in CL the auxiliary model and the target model are models in the same track. Moreover, in CL the auxiliary model is stable across the learning procedure, while itself needs careful learning in GAN.

It's worth noting that although our CL method bears certain level of resemblance with Noise Contrastive Estimation (NCE) [5]. The motivation and the actual technical formulation of CL and NCE are essentially different. For example, in NCE the logistic function is a result of computing posterior probabilities, while in CL it is explicitly introduced to saturate the influence of easy samples.

As CL requires only $p_m(c|I)$ and $p_n(c|I)$, the choices of the target model and the reference model can range from models based on LSTMs [6] to models in other formats, such as MRFs [4] and memory-networks [18]. On the other hand, although in CL, the reference model is usually fixed across the learning procedure, one can replace the reference model with the latest target model periodically. The reasons are (1) $\nabla J(\boldsymbol{\theta}) \neq \mathbf{0}$ when the target model and the reference model are identical, (2) latest target model is usually stronger than the reference model, (3) and a stronger reference model can provide stronger bounds and lead to a stronger target model.

| COCO Online Testing Server C5 | | | | | | | |
|---|---|---|---|---|---|---|---|
| Method | B-1 | B-2 | B-3 | B-4 | METEOR | ROUGE_L | CIDEr |
| Google NIC [23] | 0.713 | 0.542 | 0.407 | 0.309 | 0.254 | 0.530 | 0.943 |
| Hard-Attention[24] | 0.705 | 0.528 | 0.383 | 0.277 | 0.241 | 0.516 | 0.865 |
| AdaptiveAttention [15] | 0.735 | 0.569 | 0.429 | 0.323 | 0.258 | 0.541 | 1.001 |
| AdpativeAttention + CL (Ours) | 0.742 | 0.577 | 0.436 | 0.326 | **0.260** | 0.544 | 1.010 |
| PG-BCMR [14] | **0.754** | **0.591** | **0.445** | **0.332** | 0.257 | **0.550** | **1.013** |
| ATT-FCN[†] [26] | 0.731 | 0.565 | 0.424 | 0.316 | 0.250 | 0.535 | 0.943 |
| MSM[†] [25] | 0.739 | 0.575 | 0.436 | 0.330 | 0.256 | 0.542 | 0.984 |
| AdaptiveAttention[†] [15] | 0.746 | 0.582 | 0.443 | 0.335 | 0.264 | 0.550 | 1.037 |
| Att2in[†] [19] | - | - | - | 0.344 | 0.268 | 0.559 | 1.123 |
| COCO Online Testing Server C40 | | | | | | | |
| Method | B-1 | B-2 | B-3 | B-4 | METEOR | ROUGE_L | CIDEr |
| Google NIC [23] | 0.895 | 0.802 | 0.694 | 0.587 | 0.346 | 0.682 | 0.946 |
| Hard-Attention [24] | 0.881 | 0.779 | 0.658 | 0.537 | 0.322 | 0.654 | 0.893 |
| AdaptiveAttention [15] | 0.906 | 0.823 | 0.717 | 0.607 | 0.347 | 0.689 | 1.004 |
| AdaptiveAttention + CL (Ours) | **0.910** | **0.831** | **0.728** | **0.617** | **0.350** | **0.695** | **1.029** |
| PG-BCMR [14] | - | - | - | - | - | - | - |
| ATT-FCN[†] [26] | 0.900 | 0.815 | 0.709 | 0.599 | 0.335 | 0.682 | 0.958 |
| MSM[†] [25] | 0.919 | 0.842 | 0.740 | 0.632 | 0.350 | 0.700 | 1.003 |
| AdaptiveAttention[†] [15] | 0.918 | 0.842 | 0.740 | 0.633 | 0.359 | 0.706 | 1.051 |
| Att2in[†] [19] | - | - | - | - | - | - | - |

Table 2: This table lists published results of state-of-the-art image captioning models on the online COCO testing server. † indicates ensemble model. "-" indicates not reported. In this table, CL improves the base model (AdaptiveAttention [15]) to gain the best results among all single models on C40.

## 5 Experiment

### 5.1 Datasets

We use two large scale datasets to test our contrastive learning method. The first dataset is MSCOCO [13], which contains $122,585$ images for training and validation. Each image in MSCOCO has $5$ human annotated captions. Following splits in [15], we reserved $2,000$ images for validation. A more challenging dataset, InstaPIC-1.1M [18], is used as the second dataset, which contains $648,761$ images for training, and $5,000$ images for testing. The images and their ground-truth captions are acquired from Instagram, where people post images with related descriptions. Each image in InstaPIC-1.1M is paired with $1$ caption. This dataset is challenging, as its captions are natural posts with varying formats. In practice, we reserved $2,000$ images from the training set for validation.

On both datasets, non-alphabet characters except emojis are removed, and alphabet characters are converted to lowercases. Words and emojis that appeared less than $5$ times are replaced with *UNK*. And all captions are truncated to have at most $18$ words and emojis. As a result, we obtained a vocabulary of size $9,567$ on MSCOCO, and a vocabulary of size $22,886$ on InstaPIC-1.1M.

### 5.2 Settings

To study the generalization ability of proposed CL method, we tested it on two different image captioning models, namely **Neuraltalk2** [8] and **AdaptiveAttention** [15]. Both models are based on *encoder-and-decoder* [23], where no attention mechanism is used in the former, and an adaptive attention component is used in the latter.

For both models, we have pretrained them by MLE, and use the pretrain checkpoints as initializations. In all experiments except for the experiment on model choices, we choose the same model and use the same initialization for target model and reference model. In all our experiments, we fixed the learning rate to be $1e$-6 for all components, and used Adam optimizer. Seven evaluation metrics have been selected to compare the performances of different models, including Bleu-1,2,3,4 [17], Meteor [11], Rouge [12] and Cider [22]. All experiments for ablation studies are conducted on the validation set of MSCOCO.

| | | | | |
|---|---|---|---|---|
| | 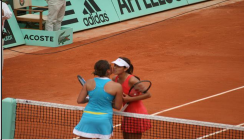 |  |  |  |
| AA | Two people on a tennis court playing tennis | A fighter jet flying through a blue sky | A row of boats on a river near a river | A bathroom with a toilet and a sink |
| AA + CL | Two tennis players shaking hands on a tennis court | A fighter jet flying over a lush green field | A row of boats docked in a river | A bathroom with a red toilet and red walls |
| |  |  |  |  |
| AA | Three clocks are mounted to the side of a building | Two people on a yellow yellow and yellow motorcycle | A baseball player pitching a ball on top of a field | A bunch of lights hanging from a ceiling |
| AA + CL | Three three clocks with three different time zones | Two people riding a yellow motorcycle in a forest | A baseball game in progress with pitcher throwing the ball | A bunch of baseballs bats hanging from a ceiling |

Figure 2: This figure illustrates several images with captions generated by different models, where *AA* represents AdaptiveAttention [15] learned by MLE, and *AA + CL* represents the same model learned by CL. Compared to *AA*, *AA + CL* generated more distinctive captions for these images.

| Method | B-1 | B-2 | B-3 | B-4 | METEOR | ROUGE_L | CIDEr |
|---|---|---|---|---|---|---|---|
| Google NIC [23] | 0.055 | 0.019 | 0.007 | 0.003 | 0.038 | 0.081 | 0.004 |
| Hard-Attention [24] | 0.106 | 0.015 | 0.000 | 0.000 | 0.026 | 0.140 | 0.049 |
| CSMN [18] | **0.079** | **0.032** | **0.015** | **0.008** | **0.037** | **0.120** | 0.133 |
| AdaptiveAttention [15] | 0.065 | 0.026 | 0.011 | 0.005 | 0.029 | 0.093 | 0.126 |
| AdaptiveAttention + CL (Ours) | 0.072 | 0.028 | 0.013 | 0.006 | 0.032 | 0.101 | **0.144** |

Table 3: This table lists results of different models on the test split of InstaPIC-1.1M [18], where CL improves the base model (AdaptiveAttention [15]) by significant margins, achieving the best result on Cider.

## 5.3 Results

**Overall Results** We compared our best model (*AdaptiveAttention* [15] learned by CL) with state-of-the-art models on two datasets. On MSCOCO, we submitted the results to the online COCO testing server. The results along with other published results are listed in Table 2. Compared to MLE-learned *AdaptiveAttention*, CL improves the performace of it by significant margins across all metrics. While most of state-of-the-art results are achieved by ensembling multiple models, our improved *AdaptiveAttention* gains competitive results as a *single* model. Specifically, on Cider, CL improves *AdaptiveAttention* from 1.003 to 1.029, which is the best single-model result on C40 among all published ones. In terms of Cider, if we use MLE, we need to combine 5 models to get 4.5% boost on C40 for *AdaptiveAttention*. Using CL, we improve the performance by 2.5% with just a single model. On InstaPIC-1.1M, CL improves the performance of *AdaptiveAttention* by 14% in terms of Cider, which is the state-of-the-art. Some qualitative results are shown in Figure 2. It's worth noting that the proposed learning method can be used with stronger base models to obtain better results without any modification.

**Compare Learning Methods** Using *AdaptiveAttention* learned by MLE as base model and initialization, we compared our CL with similar learning methods, including **CL(P)** and **CL(N)** that

| Method | B-1 | B-2 | B-3 | B-4 | METEOR | ROUGE_L | CIDEr |
|---|---|---|---|---|---|---|---|
| AdaptiveAttention [15] (Base) | 0.733 | 0.572 | 0.433 | 0.327 | 0.260 | 0.540 | 1.042 |
| Base + IL [21] | 0.706 | 0.544 | 0.408 | 0.307 | 0.253 | 0.530 | 1.004 |
| Base + GAN [1] | 0.629 | 0.437 | 0.290 | 0.190 | 0.212 | 0.458 | 0.700 |
| Base + CL(P) | 0.735 | 0.573 | 0.437 | 0.334 | 0.262 | 0.545 | 1.059 |
| Base + CL(N) | 0.539 | 0.411 | 0.299 | 0.212 | 0.246 | 0.479 | 0.603 |
| Base + CL(Full) | **0.755** | **0.598** | **0.460** | **0.353** | **0.271** | **0.559** | **1.142** |

Table 4: This table lists results of a model learned by different methods. The best result is obtained by the one learned with full CL, containing both the positive constraint and negative constraint.

| Target Model | Reference Model | B-1 | B-2 | B-3 | B-4 | METEOR | ROUGE_L | CIDEr |
|:---:|:---:|:---:|:---:|:---:|:---:|:---:|:---:|:---:|
| NT | - | 0.697 | 0.525 | 0.389 | 0.291 | 0.238 | 0.516 | 0.882 |
| NT | NT | 0.708 | 0.536 | 0.399 | 0.300 | 0.242 | 0.524 | 0.905 |
| NT | AA | **0.716** | **0.547** | **0.411** | **0.311** | **0.249** | **0.533** | **0.956** |
| AA | - | 0.733 | 0.572 | 0.433 | 0.327 | 0.260 | 0.540 | 1.042 |
| AA | AA | **0.755** | **0.598** | **0.460** | **0.353** | **0.271** | **0.559** | **1.142** |

Table 5: This table lists results of different model choices on MSCOCO. In this table, NT represents Neuraltalk2 [8], and AA represents AdaptiveAttention [15]. "-" indicates the target model is learned using MLE.

| Run | B-1 | B-2 | B-3 | B-4 | METEOR | ROUGE_L | CIDEr |
|:---:|:---:|:---:|:---:|:---:|:---:|:---:|:---:|
| 0 | 0.733 | 0.572 | 0.433 | 0.327 | 0.260 | 0.540 | 1.042 |
| 1 | 0.755 | 0.598 | 0.460 | 0.353 | 0.271 | 0.559 | 1.142 |
| 2 | 0.756 | 0.598 | 0.460 | 0.353 | 0.272 | 0.559 | 1.142 |

Table 6: This table lists results of periodical replacement of the reference in CL. In run 0, the model is learned by MLE, which are used as both the target and the reference in run 1. In run 2, the reference is replaced with the best target in run 1.

respectively contains only the positive constraint and the negative constraint in CL. We also compared with **IL** [21], and **GAN** [1]. The results on MSCOCO are listed in Table 4, where (1) among IL, CL and GAN, CL improves performance of the base model, while both IL and GAN decrease the results. This indicates the trade-off between learning distinctiveness and maintaining overall performance is not well settled in IL and GAN. (2) comparing models learned by CL(P), CL(N) and CL, we found using the positive constraint or the negative constraint alone is not sufficient, as only one source of guidance is provided. While CL(P) gives the base model lower improvement than full CL, CL(N) downgrades the base model, indicating overfits on distinctiveness. Combining CL(P) and CL(N), CL is able to encourage distinctiveness while also emphasizing on overall performance, resulting in largest improvements on all metrics.

**Compare Model Choices**  To study the generalization ability of CL, *AdaptiveAttention* and *Neuraltalk2* are respectively chosen as both the target and the reference in CL. In addition, *AdaptiveAttention* learned by MLE, as a better model, is chosen to be the reference, for *Neuraltalk2*. The results are listed in Table 5, where compared to models learned by MLE, both *AdaptiveAttention* and *Neuraltalk2* are improved after learning using CL. For example, on Cider, *AdaptiveAttention* improves from 1.042 to 1.142, and *Neuraltalk2* improves from 0.882 to 0.905. Moreover, by using a stronger model, *AdaptiveAttention*, as the reference, *Neuraltalk2* improves further from 0.905 to 0.956, which indicates stronger references empirically provide tighter bounds on both the positive constraint and the negative constraint.

**Reference Replacement**  As discussed in sec 4.3, one can periodically replace the reference with latest best target model, to further improve the performance. In our study, using *AdaptiveAttention* learned by MLE as a start, each run we fix the reference model util the target saturates its performance on the validation set, then we replace the reference with latest best target model and rerun the learning. As listed in Table 6, in second run, the relative improvements of the target model is incremental, compared to its improvement in the first run. Therefore, when learning a model using CL, with a sufficiently strong reference, the improvement is usually saturated in the first run, and there is no need, in terms of overall performance, to replace the reference multiple times.

## 6   Conclusion

In this paper, we propose Contrastive Learning, a new learning method for image captioning. By employing a state-of-the-art model as a reference, the proposed method is able to maintain the optimality of the target model, while encouraging it to learn from distinctiveness, which is an important property of high quality captions. On two challenging datasets, namely MSCOCO and InstaPIC-1.1M, the proposed method improves the target model by significant margins, and gains state-of-the-art results across multiple metrics. On comparative studies, the proposed method extends well to models with different structures, which clearly shows its generalization ability.

**Acknowledgment**   This work is partially supported by the Big Data Collaboration Research grant from SenseTime Group (CUHK Agreement No.TS1610626), the General Research Fund (GRF) of Hong Kong (No.14236516) and the Early Career Scheme (ECS) of Hong Kong (No.24204215).

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
