[Reviews · NeurIPS 2017]

Reviewer 1



The proposed method is an algorithm called Contrastive Learning used for image captioning. It's base idea is akin to use of intra/inter cluster similarity concept in clustering. The intended goal is to add "distinctiveness" to captions generated by current methods. The paper is well organized, clear in the presentation of the proposed method and in presenting the rationale for each of its aspects (section 4.2). It also draws a comparison with similar methods (Introspective Learning, Generative adversarial Networks) highlighting their main differences. Emphasis of the proposed method on enhancing caption distinctiveness through learning "negative" pairs while maintaining overall performance is also made clear. The results presented on MSCOCO and InstaPIC datasets show promising improvements. NB: Do not seem to locate the supplemental materials mentioned in paragraph 5.2.

Reviewer 2



The paper proposed a contrastive learning approach for image captioning models. Typical image captioning models utilize log-likelihood criteria for learning, which tends to result in preferring a safer generation that lacks specific and distinct concept in an image. The paper proposes to introduce contrastive learning objective, where the objective function is based on density ratio to the reference, without altering the captioning models. The paper evaluates multiple models in MSCOCO and InstaPIC datasets, and demonstrates the effectiveness as well as conducts ablation studies. The paper is well-written and has strength in the following points. * Proposing a generalizable learning method * Convincing empirical study The paper could be improved on the following respect. * Results might look insignificant depending on how to interpret * Insufficient discussion on distinctiveness vs. human-like description For the purpose of introducing distinctiveness in the image captioning problem, the paper considers altering the learning approach using existing models. The paper takes contrastive learning ideas from NCE [5], and derives an objective function eq (10). By focusing on the contrastive component in the objective, the paper solves the problem of learning under MLE scheme that results in a generic description. Although the proposed objective is similar to NCE, the approach is general and can benefit in other problem domains. This is certainly a technical contribution. In addition, the paper conducts a thorough empirical study to show the effectiveness as well as to generalization across base models and datasets. Although the result is not necessarily the best all the time depending on the evaluation scenario, I would point out the proposed approach is independent of the base model yet consistently improving the performance over the MLE baseline. One thing I would point out is that the paper could discuss more on the nature of distinctiveness in image captions. As discussed in the introduction, distinctiveness is certainly one component overlooked in caption generation. However, the paper’s view on distinctiveness is something that can be resolvable by algorithms. I would like to argue that the nature of caption data can induce generic description due to the vagueness of image content [Jas 2015]. Even if an image can be described by distinctive phrases, people can describe the content by comfortable words that are not always distinctive [Ordonez 2013]. In this respect, I would say the choice of the dataset may or may not be appropriate for evaluating distinctive phrases. The paper can add more on data statistics and human behavior on image description. * Jas, Mainak, and Devi Parikh. "Image specificity." CVPR 2015. * Ordonez, Vicente, et al. "From large scale image categorization to entry-level categories." ICCV 2013. Another concern is that image captioning is extensively studied in the past and unfortunately the paper might not be able to impact a lot in the community. In overall, the paper is well written and presenting convincing results for the concerned problem. Some people might not like the small improvements in the performance, but I believe the result also indicates a good generalization ability. I think the paper is above the borderline.

Reviewer 3



In Table 1, there are missing two important comparisons, AdaptiveAttention + IL and AdaptiveAtention + GAN. In lines 138-139 it is claimed positive correlations between self retrieval and captioning metrics, however 3 experiments are clearly not enough to claim that, it would need dozens and a proper statistical analysis. According to Table 2 the results of the proposed method are equal or just marginally better than the baseline [14], please clarify why. In Table 2, there is missing row PG-BCMR [13] which has better results on C40 that proposed method (see https://competitions.codalab.org/competitions/3221#results), and several other entries clearly better. As mentioned in [1],[3] improving on captioning metrics (BLUE, Meteor, Rouge, Cider, Spider, ...) doesn't translate into better captions when evaluated by humans. Therefore the paper would be much more convincing if they carried human evaluations. Are the numbers reported in Table 4 computed on the validation set? If so please clarify it. The paper mentions supplemental materials, but those are missing.